# Adoption of K-means clustering algorithm in smart city security analysis and mythical experience analysis of urban image

Haotong Han [1,2]*

1 Institute for Mythological Studies, Shanghai Jiaotong University, Shanghai, China, 2 School of Chinese Language and Literature, Shaanxi Normal University, Xi'an, China

* haotong665@outlook.com

**Data availability statement:** All relevant data are within the manuscript.

## Abstract

### Objective

An information security evaluation model based on the K-Means Clustering (KMC) + Decision Tree (DT) algorithm is constructed, aiming to assess its value in evaluating smart city (SC) security. Additionally, the impact of SCs on individuals' mythical experiences is investigated.

### Methods

An information security analysis model based on the combination of KMC and DT algorithms is established. A total of 38 SCs are selected as the research objects for practical analysis. The practical feasibility of the model is assessed using the receiver operating characteristic (ROC) curve, and its performance is compared with that of the Naive Bayes (NB), Logistic Regression (LR), Random Forest (RF), Support Vector Machine (SVM), and Gradient Boosting Machine (GBM) classification methods. Lastly, a questionnaire survey is conducted to obtain and analyze individuals' mythical experiences in SCs.

### Results

(1) The area under the ROC curve is significantly higher than 0.9 (0.921 vs. 0.9). (2) Compared to the NB and LR algorithms, the security analysis model based on the combination of KMC and DT algorithms demonstrated higher true positive rate (TPR), accuracy, recall, F-Score, AUC-ROC, and AUC-PR. Additionally, the performance metrics of RF, SVM, and GBM are similar to those of the KMC+DT model. (3) When the attributes are the same, the difference in smart risk levels is small, while when the attributes are different, the difference in risk levels is significant. (4) The support rates for various types of new folk activities are as follows: offline shopping festivals (17.6%), New Year's Eve celebrations (16.7%), Tibet tourism (15.6%), spiritual practices (16.2%), green leisure (16.0%), and suburban/rural tourism (15.8%). (5) High-risk cities (Grade A) showed stronger support for modern activities such as offline shopping festivals and green leisure, while low-risk cities (Grades C and D) tended to favor traditional cultural activities.

**Funding:** The author(s) received no specific funding for this work.

**Competing interests:** No authors have competing interests.

## Conclusion

The algorithm model constructed in this work is capable of effectively evaluating the information security risks of SCs and has practical value. A good city image and mythological experience are driving the development of cities.

## Introduction

With the widespread application of the Internet of Things (IoT), cloud computing, and the latest information technologies, urban management has become more refined and dynamic, giving rise to smart cities (SC) that are centered around information resources [1,2]. The development of SCs has brought significant benefits in terms of economic growth, social harmony, and environmental sustainability [3]. In recent years, the rapid development of SCs has become increasingly reliant on information technology, showing an irreversible trend [4,5]. However, alongside the increasing level of informatization, SCs are also facing severe challenges in information security. Information security issues have profound implications for urban infrastructure (such as urban safety, social security, and government security) and individual privacy, including problems like inadequate data quality, sensitive data breaches, and external malicious theft [6]. Therefore, to address these security threats, it is crucial to establish a scientific and rational information security risk assessment method.

Currently, deep learning models such as intelligent healthcare and facial recognition have demonstrated significant potential in various applications within SCs, but they still face challenges [7,8]. The literature indicates that unsupervised methods based on convolutional neural networks (CNNs) have shown advantages in visual pattern classification [9,10], yet how to effectively apply these methods to risk assessment in SCs requires further investigation. For example, some studies have proposed risk assessment models that combine Ward clustering and C4.5 Decision Tree (DT) classification [11]. C4.5 is a classification algorithm in machine learning that selects attributes based on information gain ratio and performs pruning during the DT construction process. It is well-suited for handling non-discrete and incomplete data, offering high classification accuracy. Comparisons show that while Ward clustering is more convenient for classifying complex data information, the K-means clustering (KMC) algorithm offers advantages in terms of analysis speed and complexity [12]. Currently, KMC-based information security risk assessment methods have shown promising applications in areas such as traffic information and card information security [13,14].

In practical applications, cities with similar economic and technological backgrounds, social security standards, and cultural development stages tend to exhibit similar risk levels, whereas cities with significant differences in these attributes show higher or lower variations in risk levels. For instance, factors such as cultural background, technological maturity, data privacy protection, and urban governance models can significantly influence a city's risk level [15]. On one hand, cities with advanced technology and a strong emphasis on privacy protection have better capabilities in managing security threats, resulting in relatively lower risk levels. On the other hand, cities with weaker privacy awareness and relatively underdeveloped technology applications are more prone to data breaches and malicious attacks, leading to an increase in risk levels. These attribute differences are validated through multivariate data analysis, with results showing significant disparities in security assessments across cities under the context of informatization, thereby influencing the formulation of risk management strategies.

Moreover, the construction of SCs relies not only on technological development but also on the cultivation of urban culture. In recent years, many cities in China have strengthened

the development of folk culture, encompassing both traditional cultural practices and emerging cultures derived from urban development [16,17]. Cultural factors play a crucial role in enhancing a city's image. Some scholars have pointed out that experiences such as the "urban myth" in folk activities, including shopping festivals and celebratory events, contribute to the construction of urban identity, while green leisure activities promote the purification of the urban environment. In the context of SCs, further exploration of the social effects brought about by cultural experiences has become a key area of research.

Based on the aforementioned background, this study develops an information security assessment model combining the KMC algorithm and the C4.5 DT classification method, and validates its effectiveness through real-world data. Additionally, a questionnaire survey is conducted to analyze the impact of life interactions in SCs on cultural experiences. This research aims to provide a scientific approach for information security assessment in SCs, while also enhancing our understanding of the advantages and disadvantages of cultural experiences within these urban environments.

In summary, this paper develops an information security risk assessment model through the integration of the KMC algorithm and C4.5 DT classification method. The model covers 38 cities with varying levels of SC development and has been validated through a comparison of real-world data. Furthermore, this study analyzes the cultural experiences within life interactions in SCs. This research not only offers a new method for information security management in SCs but also broadens our understanding of cultural experiences in these urban settings.

## Materials and methods

### 2.1. Model construction of information security assessment

The information security assessment model here is the C4.5 DT assessment model under KMC. It is mainly composed of three parts. The first part is data pre-processing, that is, the unification of data attributes. The second part is data pre-classification under KMC, which is the classification of data with the same attributes. The third part refers to data set classification with the C4.5 DT. That is, the DT is built to classify and organize specific data of the same attributes. The specific processing methods are as follows.

**Data pre-processing.** In the information security risk assessment process of SCs, many of the involved indicators are difficult to quantify (*e.g.,* security management awareness, risk perception). Therefore, the Delphi method must be employed to gather expert opinions and determine appropriate evaluation values for these non-quantifiable indicators (Fig 1). The evaluation values for these non-quantifiable indicators are typically collected through expert questionnaires, where experts provide scores based on their experience and understanding of urban information security, thereby supplementing the missing quantitative data. Furthermore, standardization is essential to address the issue of incomparability between indicators caused by differences in dimensions and units of measurement.

Standardization method is as follows. For quantifiable indicators (*e.g.,* device vulnerabilities, information content), it is necessary to standardize the raw data to allow for comparison across indicators with different scales and dimensions. The core of the standardization method is to transform the raw data into a common unit of measurement (*e.g.,* a scale from 0 to 10), facilitating comprehensive comparison and analysis of different indicators. For cost-related indicators, a min-max standardization method is used, where each data point is subtracted by the minimum value of the column and then divided by the difference between the maximum and minimum values, ensuring that the data falls within a uniform range. For benefit-related indicators, a similar standardization method is applied, but in this case, higher

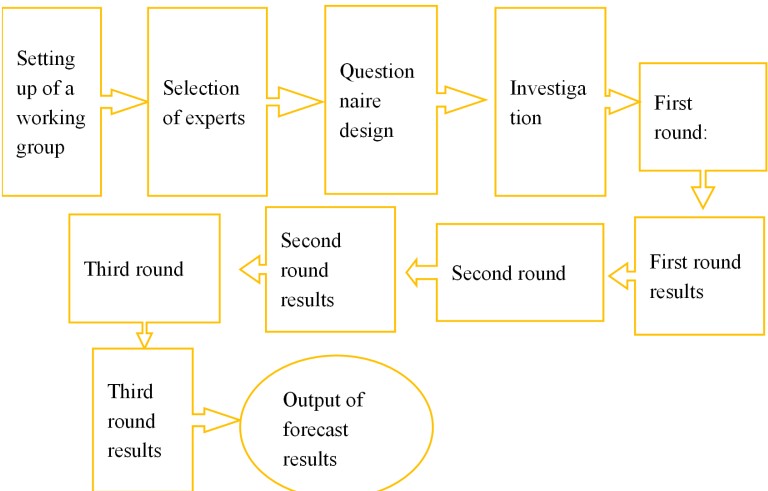

**Fig 1. Flow chart of the Delphi method.**

values indicate greater benefits. The study demonstrates the standardization process by taking both cost and benefit attribute indicator data as examples.

It is assumed that the data set is $M$, and the number of samples is $t$. The value of the data attribute $E_z$ is expressed as $R_{tz}$, the maximum value under the $E$ attribute is $\max(R_t)$, and the corresponding minimum value is $\min(R_t)$. After that, standardization processing is carried out using the specific method as Equation 1 below.

$$\overline{R_{tz}} = \frac{\left(100 \times \max(R_t) - 100 \times R_{tz}\right)}{\left(\max(R_t) - \min(R_t)\right)} \tag{1}$$

The above equation shows the standardization processing of cost-attribute indicator data, where $\overline{R_{tz}}$ represents the value after standardization processing. Equation 2 is the standardization processing of benefit-attribute indicator data, where $\overline{R_{tz}}$ represents the value after the standardization processing as well.

$$\overline{R_{tz}} = \frac{\left(100 \times R_{tz} - 100 \times \min(R_t)\right)}{\left(\max(R_t) - \min(R_t)\right)} \tag{2}$$

**Data pre-classification under KMC.** By relevant literatures are looked up, there are 20 indicators that can evaluate information security in the current information security risk assessment in SC. These indicators are information equipment and systems (F1), data hard disks and links (F2), information content resources (F3), information managers (F4), public security awareness (F5), equipment security vulnerabilities (F6), data carrier vulnerabilities (F7), information content blind area (F8), misunderstanding of application business (F9), weak area of security awareness (F10), natural man-made physical threats (F11), leakage and damage threats (F12), guidance and control threats (F13), wrong operation threats (F14), social environmental threats (F15), security inspection (F16), data isolation and encryption (F17), content supervision system (F18), security management system (F19), and security knowledge promotion (F20). All these above are condition-attribute indicators. However, due to the lack of class attribute, the DT method can't be performed normally. Therefore, it is necessary to add the 21st indicator as a categorical attribute to ensure that the DT method

can differentiate between the information risk levels of different cities. Subsequently, all raw data are clustered according to the attributes of various risk indicators. Given that the KMC model outperforms other clustering methods in terms of execution speed and complexity, and since it is an unsupervised clustering method that is easy to implement and yields good clustering results, it has been widely applied. Therefore, in this study, KMC is used for data classification.

The principle of KMC for data classification follows cost minimization. The objective function - positive real number function is obtained from the mapping from the input data and the clustering scheme. It is assumed that the objective function is $F$, then, the calculation method to find the minimum $F[(A,d),C]$ is as Equation 3. In the equation, $C$ represents the type of clustering, and $(A, d)$ stood for the input data.

$$F\Big[(A,d),\big(C_1,\cdots C_j\big)\Big]=\sum_{i=1}^{j}\int_{A\in C_i}d(A,\alpha_i(C_i))^2 \tag{3}$$

In the above equation, $j$ represents the number of clusters, $A$ is the data set, $i$ represents the data mark with the smallest distance difference from $C_i$, and $\alpha_i$ is the cluster centre. The distance between the features of different data is worked out by calculating the similarity. That is, the smaller the distance, the greater the similarity, and vice versa. Nowadays, the common classical similarity measurement methods mainly include Euclidean distance method, Pearson correlation coefficient, cosine similarity, and dynamic time warping algorithm. The Euclidean distance method provides an intuitive and easily observable approach for measuring distances in two-dimensional data. Therefore, this study employs the Euclidean distance method to calculate similarity and assist in the clustering of the data. The specific application of the Euclidean distance method is as follows.

It is assumed that there are S-dimensional features between the two data sets $P$ and Q, where $P=(p_1,p_2,\cdots,p_s)$ and $Q=(q_1,q_2,\cdots,q_s)$. Then the similarity between the two data sets is calculated as Equation 4.

$$Sim(P,Q)=\frac{P\cdot Q}{\|P\|\|Q\|} \tag{4}$$

The dot product of $P$ and $Q$ is denoted by $P\cdot Q$, and the norms of $P$ and $Q$ are denoted by $\|P\|$ and $\|Q\|$, respectively.

As it is applied to the classification of SCs, each object in the data needs to be set as a class, and the objects with the highest similarity are merged. After, the similarity between the merged class and other categories is calculated, and the above steps are repeated. The specific method is as following Equations 5 and 6.

$$Dis=\sum_{k=0}^{L_i}\overline{(C_{ik}-C_i)}\big(C_{ik}-C_i\big) \tag{5}$$

$$\overline{\overline{Dis}}=\sum_{i=0}^{j}\sum_{k=0}^{L_i}\overline{(C_{ik}-C_i)}\big(C_{ik}-C_i\big) \tag{6}$$

In the above equations, $j$ indicates that the overall data $G$ can be divided into $j$ categories, and $C_{ik}$ indicates the k-th sample in $G_i$. There are $L_i$ samples in the $G_i$ class, and $C_i$ indicates the centroid of $G_i$. $Dis$ is the sum of squared deviations of the data in the $G_i$ class, and $\overline{Dis}$

represents the sum of squared deviations within the total data class. In the process of merging clustering, it is set $G$ be at the minimum value. According to the above merging method, all cities can be merged into $N$ different clusters finally. Then, these $N$ clusters correspond to $N$ different risk levels.

**Data set classification under C4.5 DT.** After clustering with KMC, the cities are divided into different categories: $S_1$, $S_2$, $S_3$, $S_N$. However, the risk levels of each city remain unknown. At this point, the C4.5 DT method is employed to calculate the risk values and risk levels for each city. C4.5 is an algorithm based on information gain ratio that constructs a classification tree by recursively splitting the dataset to determine the information security risk levels of the cities. Compared to the traditional ID3 algorithm, C4.5 effectively avoids bias toward attributes with more attribute values and generates DTs that are intuitive and easy to understand, facilitating risk data analysis. Furthermore, C4.5 uses post-pruning techniques to remove unnecessary branches, preventing overfitting, enhancing the model's generalization ability, and ensuring the stability and accuracy of the classification results. The establishment and application of C4.5 DT can be divided into 4 steps, which are shown in Fig 2.

The specific steps are as follows.

Step 1: For discrete-continuous data sets, the information gain rate of each attribute is calculated, which is as Equation 7.

$$GainRatio(I) = \frac{Gain(I)}{SplitInfo_I(K)} \tag{7}$$

Step 2: The $MaxGainRatio(I)$ attribute is set as a branch node, and DT branches are constructed through iterations until all sample attributes are consistent.

Step 3: To prevent overfitting of the data sets, the DT needs to be post-pruned. That is, noise data and isolated nodes are eliminated.

Step 4: The DT is output, and the test data sets are classified.

**Risk assessment method.** Although the DT model can classify different information security risk classes in SC, the specific risk value must be calculated to quantify the risk level accurately. The risks caused by different risk levels are also different to the overall risk of the SC. By quantifying the impact, the weight of the overall risk corresponding to different risk levels could be expressed. That is, the $GainRatio(I)$ in the first layer of each attribute

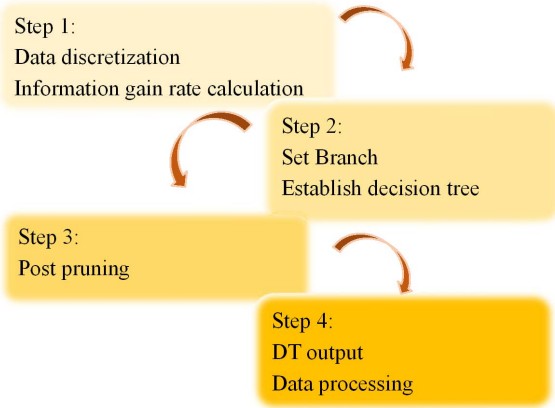

**Fig 2. C4.5 establishment process of C4.5 DT model.**

is calculated before the DT builds a branch, and the essence of *GainRatio*($I$) represents the importance of the attribute in the classification process. Then, the greater the impact of the indicator on the overall risk, the greater its weight. Therefore, in this work, the information gain rate *GainRatio*($I$) of each attribute on the root node is set as the weight, and the specific risk value of each city is obtained by weighting. The relative risk level of the city is judged relying on the obtained risk value.

## 2.2. Assessment method

To investigate the adoption advantages of the KMC-based C4.5 DT model in the information security risk assessment of SCs, this model is compared with more complex machine learning classification methods, including Naive Bayes (NB) [18], Logistic Regression (LR) [19], Random Forest (RF) [20], Support Vector Machine (SVM) [21], and Gradient Boosting Machine (GBM) [22]. The classification performance is evaluated using metrics such as true positive rate (TPR), precision, recall, F-score, the area under the precision-recall curve (PR AUC), and the area under the receiver operating characteristic (ROC) curve (AUC). The PR curve is particularly suitable for evaluating model performance in cases of imbalanced data, while the ROC curve's vertical axis represents TPR and the horizontal axis represents the False Positive Rate. The closer the curve is to the upper-left corner, the larger the AUC, indicating better model performance. In general, an AUC > 0.9 indicates high accuracy of the model.

## 2.3. Research samples

This study selects data from 38 representative SCs to validate the effectiveness of the model. Based on the *2023 China SC Development Level Assessment Report* and multiple relevant sources, 38 cities with relatively high levels of SC development are chosen as the research subjects. The selected cities include Beijing, Hangzhou, Shanghai, Shenzhen, Wuxi, Ningbo, Suzhou, Chengdu, Guangzhou, Tianjin, Nanjing, Wuhan, Chongqing, Qingdao, Zhengzhou, Shenyang, Xi'an, Xiamen, Jinan, Changsha, Luoyang, Zhuhai, Kunming, Dalian, Harbin, Changchun, Nanchang, Nanning, Hefei, Fuzhou, Guiyang, Zhangzhou, Tangshan, Lanzhou, Yangzhou, Shantou, Huaian, and Langfang. These cities are ranked according to their comprehensive scores in SC development level and informatization construction, with rankings numbered from 1 to 38. The criteria for city selection include: first, mature SC practices; second, high levels of informatization and intelligence; and third, the inclusion of municipalities directly under the central government and provincial capitals. In determining the sample size, consideration is given to ensuring representativeness by covering cities from different regions with varying levels of development. To this end, this study employs a stratified sampling approach to ensure that cities with significant SC characteristics are adequately represented in the sample. Data for the research subjects are collected using both the Delphi method and a questionnaire survey to ensure the diversity and representativeness of the sample. Specifically, in addition to the values and rankings provided in the report, expert evaluations are obtained via the Delphi method (expert questionnaire shown in Table 1) and a public questionnaire survey. The Delphi method targets experts to reduce subjective bias, while the public questionnaire aims to gauge public awareness of information security risks (with 0 indicating "unsafe" and 10 indicating "safe"). The combination of expert opinions and public perspectives further enhances the representativeness of the study sample.

## 2.4. Assessment indicators

The information security risk assessment system for SC integrates risk assessment factors and security risk levels, aiming to comprehensively evaluate and manage information security

Table 1. Questionnaire for experts on security risks of information management in SC.

| SCs | Information managers (a) | Misunderstanding of application business (b) | Threat of wrong operations (c) | Security management system (d) |
|---|---|---|---|---|
| City (1) | Dependence | Number of loopholes | Severity of consequences | Degree of perfection |
| City (2) | —— | —— | —— | —— |
| ………… | | | | |
| City (15) | —— | —— | —— | —— |
| (a): The higher the dependence on managers, the higher the risk. | | | | |
| (b): The more management loopholes appear, the higher the risk. | | | | |
| (c): The more serious the consequences, the higher the risk. | | | | |
| (d): The more imperfect the management system, the higher the risk. | | | | |
| 0 - the highest risk; 10 - the lowest risk. | | | | |

risks. The risk assessment factors include assets, threats, vulnerabilities, and security measures. This study further expands the factors to include privacy protection, emergency response capability, data integrity, access control, compliance and auditing, IoT device security, security awareness among employees and the public, and data redundancy and backup, to ensure a broader and deeper coverage of risks. The security risk levels primarily assess the security of five key areas: infrastructure security, data services, information content, application services, and public awareness. The impact of each risk assessment indicator is assigned a score between 1 and 9 and categorized into five levels: A, B, C, D, and E, from high to low. The assessment system is set such that higher values for assets, vulnerabilities, and threats indicate a higher potential risk level, while higher values for security measures (such as access control and data redundancy) and emergency response capabilities suggest stronger resilience against risks, thereby reducing the impact on the overall risk level. The specific grading standards are as follows: Level A (extremely high risk): risk score of 8-9, indicating that the city faces significant potential threats in the corresponding risk areas and lacks effective security controls. These cities have weak risk management capabilities and require priority implementation of comprehensive risk mitigation measures. Level B (high risk): risk score of 6-7, indicating that the city exhibits significant security risks in some areas. While some security measures are in place, further strengthening is needed. Focus should be placed on improving emergency response capabilities, data redundancy, and access control measures. Level C (medium risk): risk score of 4-5, indicating that the city has moderate performance in risk control, with some capacity to respond to threats. However, certain critical areas (such as IoT security and data protection) still present potential vulnerabilities. These cities can enhance their security level by strengthening specific security measures. Level D (low risk): risk score of 2-3, indicating that the city's risk prevention measures are relatively well-developed, with effective risk control, though a few minor potential threats remain. Continuous monitoring and assessment of emerging threats are needed. Level E (extremely low risk): risk score of 1, indicating that the city has strong security protection measures across all risk domains and possesses a very high response capability to potential threats. These cities have highly mature risk management systems but must remain vigilant about technological developments to prevent new threats from arising.

## 2.5. Mythical experience survey

Regarding the analysis of people's mythical experience of the city image, the citizens in 38 SCs are as the survey objects. 30 people in each city are randomly selected for the investigation of the representative new folk customs in the SC. The mythical experience of new folk customs is compared and analysed using the theory of mythology, to discuss the mythical dimension

in SC life. For the survey, the method of street interview is adopted. The specific question is "Which one do you think is the most representative new folk customs in SC? Offline shopping festivals, New Year's Eve celebrations, Tibetan travel, spiritual practice, green leisure, or suburban countryside tourism?" After that, the outcomes of the interviews are counted, and the support rate of each new folk custom was calculated. The specific calculation method is as Equation 8 below.

$$Rate_{(Support)} = \frac{N_{(Support)}}{N_{All}} \times 100\% \tag{8}$$

In the equation, $Rate_{(Support)}$ represents the support rate, $N_{(Support)}$ represents the number of people who support a folk custom, and $N_{All}$ represents the total number of people.

The study also further analyzes the support rates for new cultural practices across cities with different security risk levels, to assess the potential relationship between risk levels and cultural activities.

## Results and discussion

### Collation of experts' assessment results

Fig 3 displays the expert survey rating results for the 38 SCs, which were organized using the Delphi method. The survey consisted of three rounds of ratings, with the median of each indicator in the final round serving as the ultimate result. In this work, only the survey rating results for Beijing are presented due to the large volume of data. The results show that there are certain differences in the rating results among the indicators in each round. However, as the survey progresses, there is a decreasing trend in the standard deviation between the expert rating results.

### Questionnaire on the risk of public literacy security

In the context of SC, 30 questionnaires on public perception of security risks were distributed to residents of each of the 38 cities. Fig 4 presents the detailed results of the questionnaires from these 38 cities. A total of 1,140 questionnaires were distributed, of which 934 were

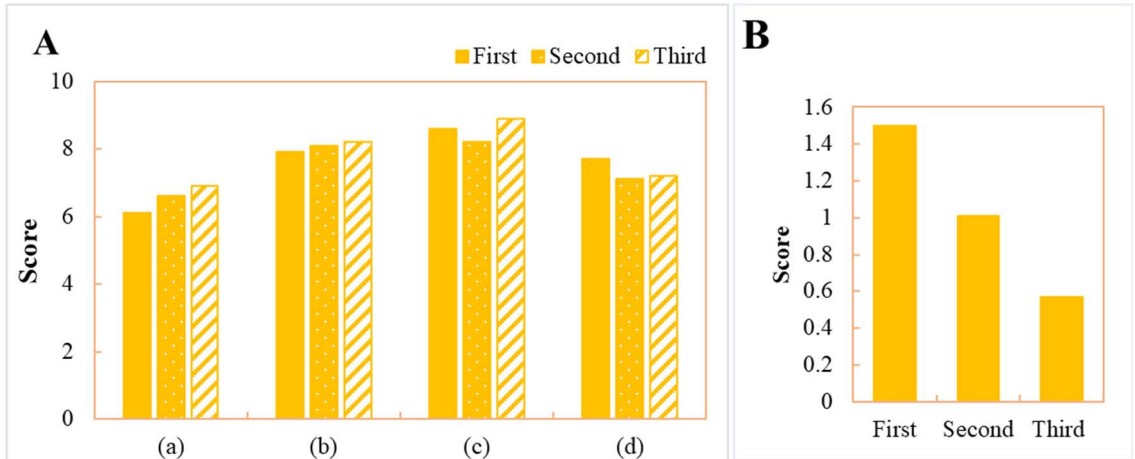

**Fig 3. Expert scoring results of Beijing's information management security risks in SC.** (A) Scoring results of the three rounds. (B) Standard deviation.

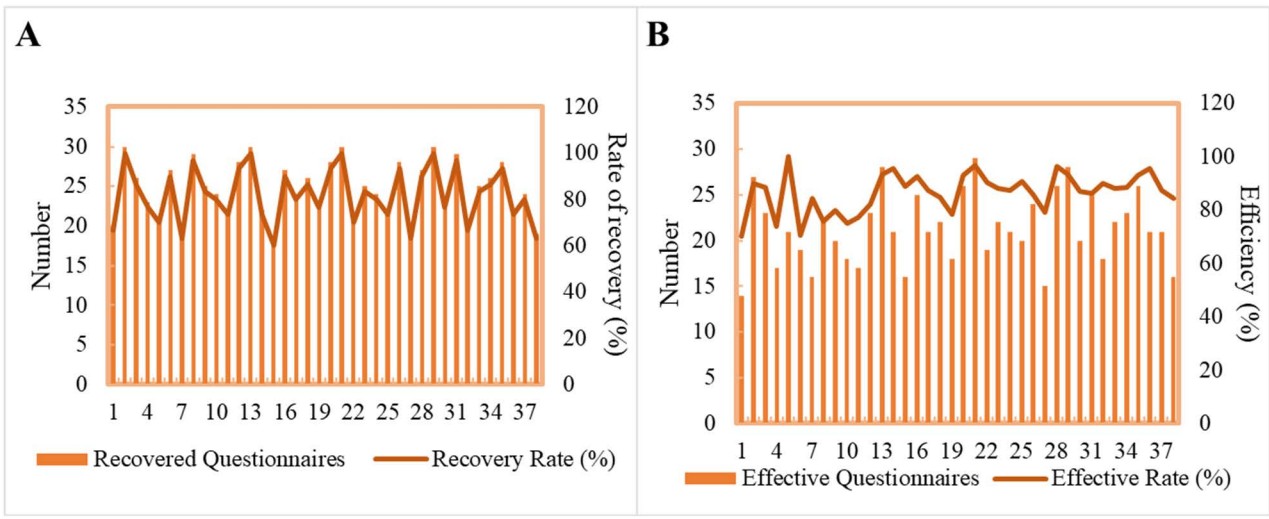

**Fig 4. The recovery (A) and effectiveness (B) of the questionnaire on the risk of public literacy security in SCs.**

returned, yielding a response rate of 81.92%. Of the returned questionnaires, 810 are valid, resulting in a validity rate of 86.53%.

## Application process of risk assessment model under KMC algorithm-based C4.5 DT

**Data consistency processing.** The 20 assessment indicators are denoted as F1-F20, respectively. Through the data pre-processing, the original survey data is processed consistently; the standardized scoring data in Table 2 are obtained.

**KMC pre-classification processing.** The study employs relative values instead of absolute values for risk level substitution. The original dataset was subjected to KMC analysis using *SPSS*. First, the dataset is standardized using Z-scores to ensure the balance and comparability of the data. Next, based on the standardized data, a dendrogram (Fig 5) is created to illustrate the distribution of different cities across various attribute categories.

Based on the KMC analysis results, the dataset is divided into five attribute categories. The specific classifications are as follows:

Attribute 1 (S1): Zhuhai, Nanchang, Zhengzhou, Fuzhou, Kunming, Dalian, Guiyang, Tangshan, Hefei.

Attribute 2 (S2): Wuhan, Suzhou, Chengdu, Nanjing, Qingdao, Zhangzhou, Yangzhou, Lanzhou.

Attribute 3 (S3): Beijing, Shenzhen, Shanghai, Guangzhou, Tianjin, Jinan, Hangzhou.

Attribute 4 (S4): Changsha, Chongqing, Shenyang, Luoyang, Harbin, Shantou, Huaian.

Attribute 5 (S5): Wuxi, Ningbo, Xi'an, Xiamen, Changchun, Nanning, Langfang.

On this basis, the original dataset was further processed according to the attribute classifications and risk level scores, providing more accurate and multidimensional risk assessment results for subsequent analysis.

With the continuous improvement and application of artificial intelligence and information systems, the number of SCs in China is increasing. Although the management of SCs

**Table 2. Consistency results of raw data.**

| City Number | F1 | F2 | F3 | F4 | F5 | F6 | F7 | F8 | F9 | F10 | F11 | F12 | F13 | F14 | F15 | F16 | F17 | F18 | F19 | F20 |
|---|---|---|---|---|---|---|---|---|---|---|---|---|---|---|---|---|---|---|---|---|
| 1 | 7.9 | 4.9 | 5.9 | 7.9 | 6.9 | 7.9 | 4.9 | 6.9 | 3.9 | 5.9 | 3 | 4.9 | 6.9 | 7.9 | 8.9 | 6.9 | 6.9 | 7.9 | 4.9 | 3.9 |
| 2 | 8.9 | 2 | 3 | 3 | 5.9 | 8.9 | 4.9 | 5.9 | 3 | 5.9 | 3.9 | 3.9 | 6.9 | 6.9 | 5.9 | 5.9 | 6.9 | 6.9 | 4.9 | 4.9 |
| 3 | 6.9 | 4.9 | 6.9 | 6.9 | 2 | 5.9 | 3.9 | 3.9 | 3 | 5.9 | 3.9 | 2 | 5.9 | 5.9 | 4.9 | 4.9 | 4.9 | 4.9 | 5.9 | 3 |
| 4 | 6.9 | 7.9 | 3.9 | 5.9 | 2 | 7.9 | 3.9 | 4.9 | 3.9 | 5.9 | 2 | 4.9 | 5.9 | 6.9 | 4.9 | 6.9 | 5.9 | 5.9 | 5.9 | 4.9 |
| 5 | 5.9 | 5.9 | 7.9 | 6.9 | 5.9 | 6.9 | 8.9 | 7.9 | 3 | 5.9 | 1 | 5.9 | 4.9 | 5.9 | 6.9 | 5.9 | 6.9 | 7.9 | 4.9 | 6.9 |
| 6 | 5.9 | 5.9 | 7.9 | 7.9 | 7.9 | 5.9 | 8.9 | 6.9 | 1 | 3.9 | 2 | 5.9 | 4.9 | 5.9 | 3.9 | 4.9 | 4.9 | 6.9 | 4.9 | 4.9 |
| 7 | 4.9 | 5.9 | 8.9 | 3.9 | 8.9 | 8.9 | 7.9 | 4.9 | 3.9 | 5.9 | 4.9 | 5.9 | 4.9 | 5.9 | 3.9 | 7.9 | 6.9 | 4.9 | 7.9 | 3 |
| 8 | 6.9 | 5.9 | 2 | 6.9 | 6.9 | 7.9 | 3 | 3.9 | 3.9 | 4.9 | 3.9 | 5.9 | 4.9 | 5.9 | 7.9 | 5.9 | 7.9 | 7.9 | 4.9 | 5.9 |
| 9 | 4.9 | 4.9 | 4.9 | 4.9 | 7.9 | 4.9 | 3.9 | 3.9 | 3.9 | 4.9 | 3.9 | 2 | 3.9 | 5.9 | 4.9 | 5.9 | 5.9 | 6.9 | 5.9 | 4.9 |
| 10 | 4.9 | 3 | 4.9 | 1 | 5.9 | 4.9 | 3.9 | 4.9 | 4.9 | 3.9 | 3.9 | 4.9 | 3.9 | 4.9 | 8.9 | 4.9 | 5.9 | 6.9 | 3.9 | 4.9 |
| 11 | 5.9 | 2 | 3 | 8.9 | 3 | 5.9 | 3 | 3.9 | 4.9 | 3.9 | 3.9 | 4.9 | 6.9 | 4.9 | 7.9 | 3 | 3.9 | 3.9 | 5.9 | 5.9 |
| 12 | 6.9 | 4.9 | 5.9 | 4.9 | 5.9 | 5.9 | 3.9 | 5.9 | 4.9 | 5.9 | 4.9 | 4.9 | 3.9 | 6.9 | 5.9 | 5.9 | 6.9 | 4.9 | 4.9 | 6.9 |
| 13 | 5.9 | 4.9 | 7.9 | 5.9 | 5.9 | 5.9 | 7.9 | 8.9 | 4.9 | 6.9 | 3.9 | 6.9 | 5.9 | 4.9 | 5.9 | 4.9 | 3.9 | 5.9 | 3 | 7.9 |
| 14 | 4.9 | 5.9 | 7.9 | 5.9 | 8.9 | 8.9 | 4.9 | 4.9 | 4.9 | 3 | 5.9 | 5.9 | 6.9 | 8.9 | 3.9 | 8.9 | 7.9 | 4.9 | 3.9 | 7.9 |
| 15 | 4.9 | 5.9 | 7.9 | 5.9 | 8.9 | 4.9 | 5.9 | 5.9 | 4.9 | 6.9 | 3 | 7.9 | 5.9 | 8.9 | 3 | 6.9 | 4.9 | 4.9 | 3.9 | 5.9 |
| 16 | 5.9 | 7.9 | 3.9 | 6.9 | 6.9 | 4.9 | 7.9 | 4.9 | 4.9 | 5.9 | 6.9 | 3 | 4.9 | 7.9 | 5.9 | 6.9 | 5.9 | 4.9 | 3.9 | 3 |
| 17 | 4.9 | 7.9 | 6.9 | 3 | 5.9 | 3.9 | 4.9 | 3.9 | 4.9 | 7.9 | 6.9 | 5.9 | 5.9 | 5.9 | 3.9 | 5.9 | 7.9 | 5.9 | 4.9 | 6.9 |
| 18 | 4.9 | 4.9 | 4.9 | 5.9 | 5.9 | 4.9 | 3.9 | 4.9 | 4.9 | 4.9 | 3.9 | 5.9 | 6.9 | 3 | 4.9 | 5.9 | 6.9 | 5.9 | 5.9 | 4.9 |
| 19 | 5.9 | 4.9 | 6.9 | 7.9 | 7.9 | 5.9 | 3.9 | 5.9 | 3 | 7.9 | 4.9 | 3.9 | 4.9 | 5.9 | 7.9 | 4.9 | 4.9 | 5.9 | 3.9 | 7.9 |
| 20 | 7.9 | 7.9 | 4.9 | 6.9 | 5.9 | 5.9 | 4.9 | 5.9 | 3.9 | 4.9 | 3.9 | 5.9 | 4.9 | 4.9 | 6.9 | 5.9 | 5.9 | 6.9 | 3.9 | 3.9 |
| 21 | 4.9 | 4.9 | 3.9 | 3.9 | 5.9 | 5.9 | 4.9 | 3.9 | 7.9 | 6.9 | 4.9 | 5.9 | 7.9 | 3.9 | 6.9 | 5.9 | 6.9 | 3.9 | 5.9 | 5.9 |
| 22 | 6.9 | 7.9 | 4.9 | 6.9 | 4.9 | 3.9 | 7.9 | 6.9 | 4.9 | 5.9 | 6.9 | 4.9 | 3.9 | 6.9 | 4.9 | 5.9 | 7.9 | 6.9 | 6.9 | 5.9 |
| 23 | 7.9 | 7.9 | 3.9 | 5.9 | 5.9 | 6.9 | 5.9 | 4.9 | 4.9 | 6.9 | 5.9 | 3.9 | 6.9 | 7.9 | 5.9 | 7.9 | 3.9 | 6.9 | 3.9 | 7.9 |
| 24 | 6.9 | 5.9 | 4.9 | 5.9 | 7.9 | 6.9 | 6.9 | 5.9 | 5.9 | 6.9 | 6.9 | 4.9 | 7.9 | 5.9 | 7.9 | 3.9 | 5.9 | 5.9 | 6.9 | 5.9 |
| 25 | 6.9 | 4.9 | 4.9 | 5.9 | 6.9 | 5.9 | 5.9 | 7.9 | 3.9 | 6.9 | 5.9 | 4.9 | 5.9 | 7.9 | 6.9 | 6.9 | 4.9 | 6.9 | 5.9 | 3.9 |
| 26 | 6.9 | 5.9 | 5.9 | 6.9 | 4.9 | 4.9 | 5.9 | 5.9 | 7.9 | 5.9 | 6.9 | 4.9 | 5.9 | 5.9 | 4.9 | 5.9 | 5.9 | 5.9 | 5.9 | 7.9 |
| 27 | 6.9 | 3.9 | 7.9 | 5.9 | 6.9 | 6.9 | 5.9 | 4.9 | 3.9 | 6.9 | 5.9 | 5.9 | 7.9 | 6.9 | 6.9 | 7.9 | 5.9 | 5.9 | 6.9 | 5.9 |
| 28 | 4.9 | 6.9 | 4.9 | 6.9 | 6.9 | 6.9 | 4.9 | 5.9 | 4.9 | 6.9 | 5.9 | 4.9 | 3.9 | 6.9 | 5.9 | 4.9 | 7.9 | 6.9 | 5.9 | 4.9 |
| 29 | 7.9 | 5.9 | 7.9 | 4.9 | 4.9 | 3.9 | 4.9 | 7.9 | 6.9 | 4.9 | 5.9 | 4.9 | 5.9 | 6.9 | 7.9 | 5.9 | 3.9 | 6.9 | 3.9 | 5.9 |
| 30 | 5.9 | 5.9 | 6.9 | 4.9 | 7.9 | 5.9 | 6.9 | 3.9 | 4.9 | 6.9 | 5.9 | 4.9 | 6.9 | 6.9 | 5.9 | 7.9 | 4.9 | 7.9 | 6.9 | 4.9 |
| 31 | 6.9 | 7.9 | 6.9 | 5.9 | 7.9 | 7.9 | 5.9 | 6.9 | 5.9 | 7.9 | 7.9 | 6.9 | 6.9 | 4.9 | 6.9 | 5.9 | 5.9 | 7.9 | 6.9 | 4.9 |
| 32 | 5.9 | 6.9 | 6.9 | 7.9 | 6.9 | 6.9 | 7.9 | 5.9 | 5.9 | 5.9 | 5.9 | 4.9 | 6.9 | 6.9 | 5.9 | 5.9 | 4.9 | 5.9 | 5.9 | 7.9 |
| 33 | 4.9 | 6.9 | 7.9 | 4.9 | 6.5 | 4.9 | 3.9 | 6.9 | 5.9 | 5.9 | 4.9 | 7.9 | 4.9 | 6.9 | 4.9 | 7.9 | 3.9 | 5.9 | 5.9 | 5.9 |
| 34 | 6.9 | 4.9 | 5.9 | 5.9 | 4.9 | 5.9 | 4.9 | 4.9 | 4.9 | 7.9 | 4.9 | 5.9 | 7.9 | 6.9 | 7.9 | 6.9 | 5.9 | 6.9 | 4.9 | 6.9 |
| 35 | 5.9 | 5.9 | 4.9 | 6.9 | 7.9 | 5.9 | 6.9 | 7.9 | 4.9 | 5.9 | 5.9 | 5.9 | 4.9 | 4.9 | 5.9 | 6.9 | 7.9 | 5.9 | 7.9 | 6.9 |
| 36 | 4.9 | 7.9 | 5.9 | 4.9 | 4.9 | 4.9 | 6.9 | 7.9 | 5.9 | 4.9 | 6.9 | 5.9 | 6.9 | 6.9 | 5.9 | 6.9 | 6.9 | 5.9 | 6.9 | 7.9 |
| 37 | 6.9 | 7.9 | 5.9 | 4.9 | 5.9 | 7.9 | 7.9 | 5.9 | 4.9 | 5.9 | 4.9 | 4.9 | 5.9 | 6.9 | 5.9 | 4.9 | 7.9 | 7.9 | 4.9 | 5.9 |
| 38 | 7.9 | 5.9 | 6.9 | 6.9 | 7.9 | 5.9 | 5.9 | 6.9 | 4.9 | 6.9 | 6.9 | 5.9 | 5.9 | 7.9 | 5.9 | 5.9 | 6.9 | 7.9 | 7.9 | 5.9 |

has been significantly improved, there is a high dependence on artificial intelligence and information systems, which inevitably brings about the risk of information security incidents. Moreover, most experts have emphasized the need for information security assurance in the development of SCs. Based on this, an information security evaluation model is established using a combination of KMC and C4.5 DT classification. Practical risk analysis is conducted, and life interactions in SCs that contribute to mythical experiences are analyzed through a questionnaire survey.

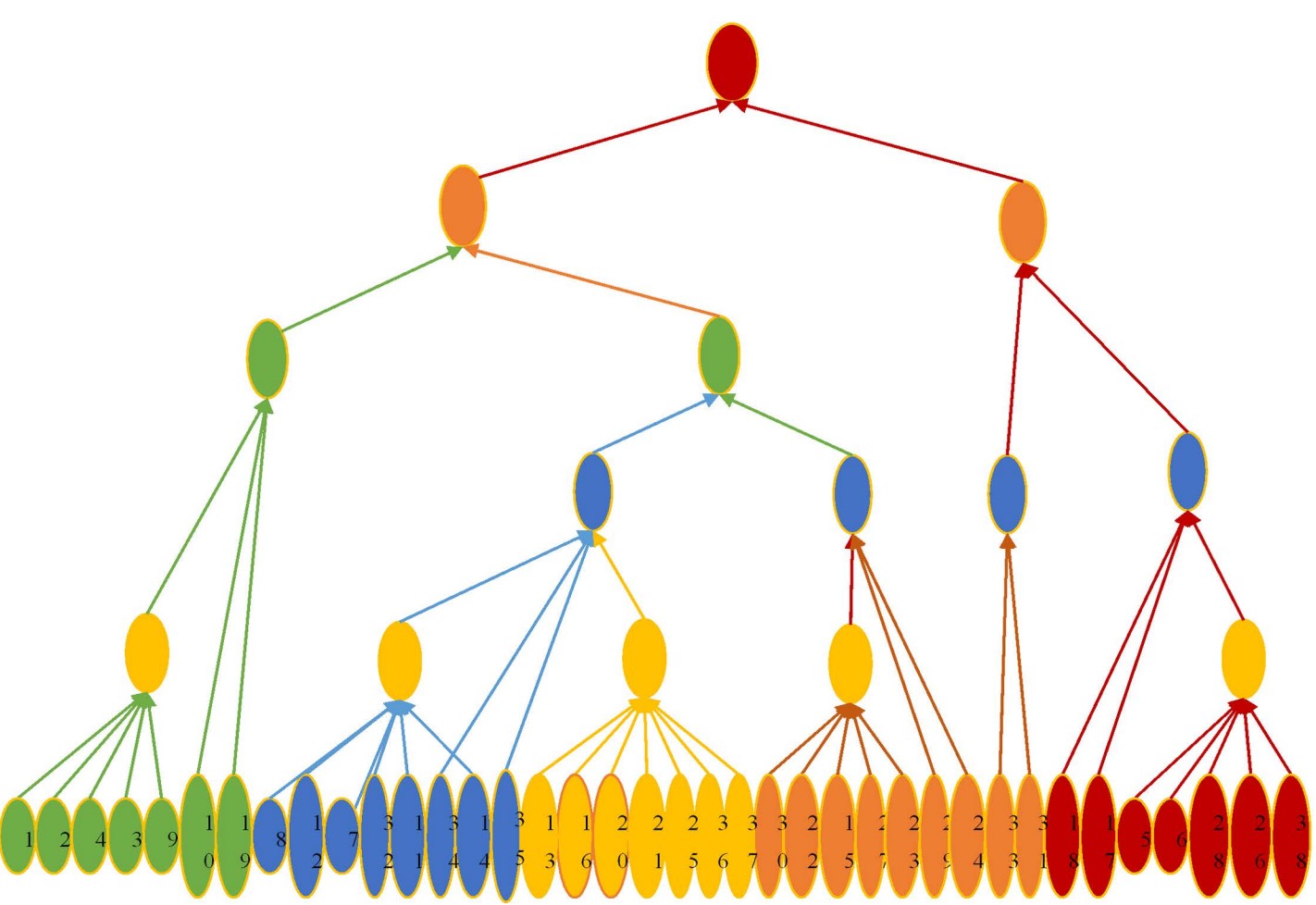

**Fig 5. KMC hierarchical diagram.**

Firstly, the Delphi method is adopted to obtain effective results for expert survey ratings of 38 SCs. Currently, the Delphi method has been widely applied and researched in various fields, including emergency nursing and pharmaceutical education [23]. Additionally, Meng (2021) [24] mentioned in their study that the results obtained through the Delphi method are clear and have good practical value. The findings of this work also reveal a decreasing trend in the standard deviation of expert ratings after multiple rounds of surveys. This suggests that, as different viewpoints are understood and the controversy surrounding various indicators reduced, the ratings become more consistent. This indicates the rationality of applying the Delphi method in this work.

Furthermore, the KMC algorithm is used in conjunction with the C4.5 DT to classify and evaluate the information security risk levels of the 38 SCs. The KMC algorithm is employed to categorize the 38 SCs into five attributes: S1, S2, S3, S4, and S5. Subsequently, the C4.5 DT model is applied to assess the risk levels of each SC. By calculating the ROC curve of the KMC algorithm-based C4.5 DT, an ROC area of 0.921 is obtained, indicating the feasibility of the model in this work. The KMC algorithm has been widely applied in various fields, demonstrating its significant practical value, such as image classification [25] and disease classification [26]. Multiple studies on KMC algorithm-based classification have demonstrated its high

accuracy in assisting with dataset classification, making it a promising area for research and application [27,28].

**Establishment and advantage analysis of C4.5 DT model.** The information gain rate of each attribute in the original data has been researched and calculated. Based on the results obtained, the maximum gain rate is taken as the gain rate of the attribute until the category consistency is satisfied, and finally a DT model with 5 leaf nodes and a size of 9 is obtained. In order to explore the advantages of the C4.5 DT model based on KMC in the information security risk assessment of supply chain (SC), it is compared with the NB, LR, RF, SVM, and GBM models. Considering the small amount of data and the impact of model complexity, this study has conducted a detailed evaluation of overfitting of the model. Firstly, during the training process, the 10-fold cross-validation method was used, in which the data was divided into 10 parts each time, with 9 parts used for training and 1 part for testing, and this process was repeated 10 times, with the results finally averaged. Cross-validation can ensure the stability of the model on different data subsets and reduce the impact caused by the randomness of data division, especially suitable for cases with a small number of samples. The validation results of each fold show that the model performs consistently on different training and test sets, with the AUC value stably maintained between 0.91 and 0.93 in multiple folds, indicating that the model has good generalization ability. To further assess the possibility of overfitting, the performance difference of the model on the training set and the test set was also calculated. The results show that the difference in AUC values on the training set and the test set is small, indicating that the model has not overfitted the training set data. Especially with a low false positive rate (0.029), it can be concluded that the classification results of the model are quite stable. In addition, regularization techniques (such as pruning methods) have been used to prevent overfitting, which helps to reduce the complexity of the DT in the training process, ensure the appropriate depth of the tree, and thus improve the generalization ability of the model (Table 3). Through these measures, it can be confirmed that there is no obvious overfitting problem in this model.

Based on the above overfitting analysis, the model has been compared with the NB, LR, RF, SVM, and GBM models. The comparison indicators include TPR, precision, recall, F-score, AUC-PR, and AUC-ROC. Table 4 shows the performance of the C4.5 DT model based on KMC in risk level classification. The results indicate that, compared with the NB and LR models, the C4.5 DT model based on KMC performs better in key performance indicators such as TPR, precision, recall, F-score, AUC-ROC, and AUC-PR, and particularly stands out in terms of false positive rate (0.029). Although the RF, SVM, and GBM models perform better in classification accuracy, they are more dependent on feature selection and parameter tuning. In contrast, the C4.5 DT model based on KMC in this study is more convenient in dealing with the complexity of feature selection and has a lower false positive rate (0.029), making it more advantageous in SC information security risk assessment. Other studies have shown that models based on ensemble methods and clustering algorithms usually perform better

**Table 3. Overfitting evaluation results.**

| Model | Training set AUC | Test set AUC | AUC difference | Overfitting evaluation | Regularization techniques |
|---|---|---|---|---|---|
| C4.5 (KMC) | 0.925 | 0.921 | 0.004 | No obvious overfitting | Pruning |
| Naive Bayes (NB) | 0.74 | 0.723 | 0.017 | No obvious overfitting | None |
| Logistic Regression (LR) | 0.895 | 0.889 | 0.006 | No obvious overfitting | L2 regularization |
| Random Forest (RF) | 0.92 | 0.91 | 0.01 | No obvious overfitting | Random forest pruning |
| Support Vector Machine (SVM) | 0.89 | 0.895 | 0.005 | No obvious overfitting | None |
| Gradient Boosting Machine (GBM) | 0.918 | 0.915 | 0.003 | No obvious overfitting | L2 regularization |

**Table 4. Comparison of performance and interpretability of different models.**

| Model | TPR | Precision | Recall | F-score | AUC-ROC | AUC-PR | False positive rate | Interpretability characteristics | Characteristics importance score | Decision path transparency | User understanding |
|---|---|---|---|---|---|---|---|---|---|---|---|
| C4.5 DT | 0.887 | 0.887 | 0.872 | 0.87 | 0.921 | 0.899 | 0.029 | Clear DT structure, easy to understand | High | High | High |
| Naive Bayes (NB) | 0.701 | 0.7 | 0.702 | 0.698 | 0.723 | 0.71 | 0.102 | Based on probability, easy to understand | Low | High | High |
| Logistic Regression (LR) | 0.802 | 0.832 | 0.802 | 0.714 | 0.889 | 0.85 | 0.062 | Simple linear relationship, easy to explain | Low | High | High |
| Random Forest (RF) | 0.875 | 0.89 | 0.867 | 0.878 | 0.91 | 0.88 | 0.03 | Multi-tree integration, feature importance can be calculated | High | Medium | Medium |
| Support Vector Machine (SVM) | 0.845 | 0.86 | 0.835 | 0.849 | 0.895 | 0.865 | 0.041 | Complex decision boundary, difficult to understand intuitively | Medium | Low | Low |
| Gradient Boosting Machine (GBM) | 0.882 | 0.892 | 0.877 | 0.885 | 0.915 | 0.892 | 0.032 | The optimization process is difficult to understand intuitively | Medium | Low | Low |

in classification accuracy than single models. Liang et al. (2021) [29] compared the selective ensemble classification model with the NB, LR, and DT classifiers in gas disaster risk identification. The results showed that the selective ensemble classification model based on clustering selection and a new degree of combination fitness (CS-NDCF) improved the classification accuracy by 34.83%, 12.94%, and 5.51%, respectively compared with the NB, LR, and DT classifiers. In particular, the accuracy of the DT classifier was significantly improved under the selective ensemble model. This finding is consistent with the results of this study, further highlighting the advantages of the C4.5 DT model based on KMC in classification performance. In addition, the combination of KMC and C4.5 DT models performs better in AUC-ROC and AUC-PR than other classification methods. Although models such as RF, SVM, and GBM perform better in classification accuracy, they are more reliant on feature selection and parameter optimization. The interpretability of the C4.5 DT model based on KMC has been further evaluated. The results show that, compared with other complex models (such as SVM and GBM), the C4.5 DT has a significant advantage in interpretability. The decision rules generated by the C4.5 DT are clear and can intuitively show how each feature affects the classification results. In contrast, although complex models such as SVM and GBM perform well in classification accuracy, their "black box" characteristics make it difficult to explain their decision-making processes. SVM classifies through support vectors, and its decision boundaries are not easy to understand intuitively; GBM optimizes the model through multiple iterations, but the logic behind each optimization is usually difficult to present. Therefore, although these models have high precision, they have poor interpretability, which may bring transparency and trust issues in practical applications. In the field of information security, especially in the information security risk assessment of SC, decision-makers need to clearly understand how each risk factor affects the overall risk assessment results. The C4.5 DT model based on KMC can provide clear rules and paths, enabling decision-makers to trace the specific basis of each risk assessment, thereby enhancing the transparency and credibility of the decision-making process. In addition, the interpretability of the model enhances the decision-makers' trust in the model, making it more advantageous in practical applications. Through feature importance analysis, this study has identified key features affecting the information security risk assessment of SC, such as "economic development level", "information construction", and "management standards", and the ranking of the importance of these features helps to understand which factors have a decisive impact on the risk assessment results. In general, the C4.5 DT model based on

KMC not only provides high classification accuracy but also maintains good interpretability, making it suitable for fields that require high transparency and trust. The study has confirmed that the C4.5 DT model based on KMC shows greater simplicity in feature selection, and has a lower false positive rate (0.029), making it more advantageous in the information security risk assessment of SC. This result is consistent with the excellent performance of the KMC combined with DT algorithm in various application scenarios reported in the literature [30,31], further confirming the potential of the C4.5 DT model based on KMC in practical applications. Therefore, the C4.5 DT model based on KMC has found a good balance between performance and interpretability, and is an ideal choice for the information security risk assessment of SC.

## Assessment results of risk value and risk level

After computation and evaluation, the risk values and risk level distribution of the 38 SCs are presented in Fig 6, where A, B, C, D, and E represent risk levels from high to low. It can be observed that cities with the same attribute exhibit minor differences, while cities with different risk attributes show significant variations in risk values and risk levels. The results indicate that the specific attributes of a city have a significant impact on its information security risk level. Specifically, these attributes include factors such as economic development level, degree of informationization, technological infrastructure, and management standards. Economically developed cities typically possess more advanced information security protection systems, resulting in lower risk levels. In contrast, cities with lower levels of informationization or weaker economies, due to underdeveloped technological infrastructure, face higher security risks. Large cities, with their larger scale and more complex operations, may encounter more sophisticated security threats, while smaller cities or those with limited resources may experience higher risks due to insufficient technological resources and lower management capabilities. These attribute differences suggest that cities should tailor their information security risk

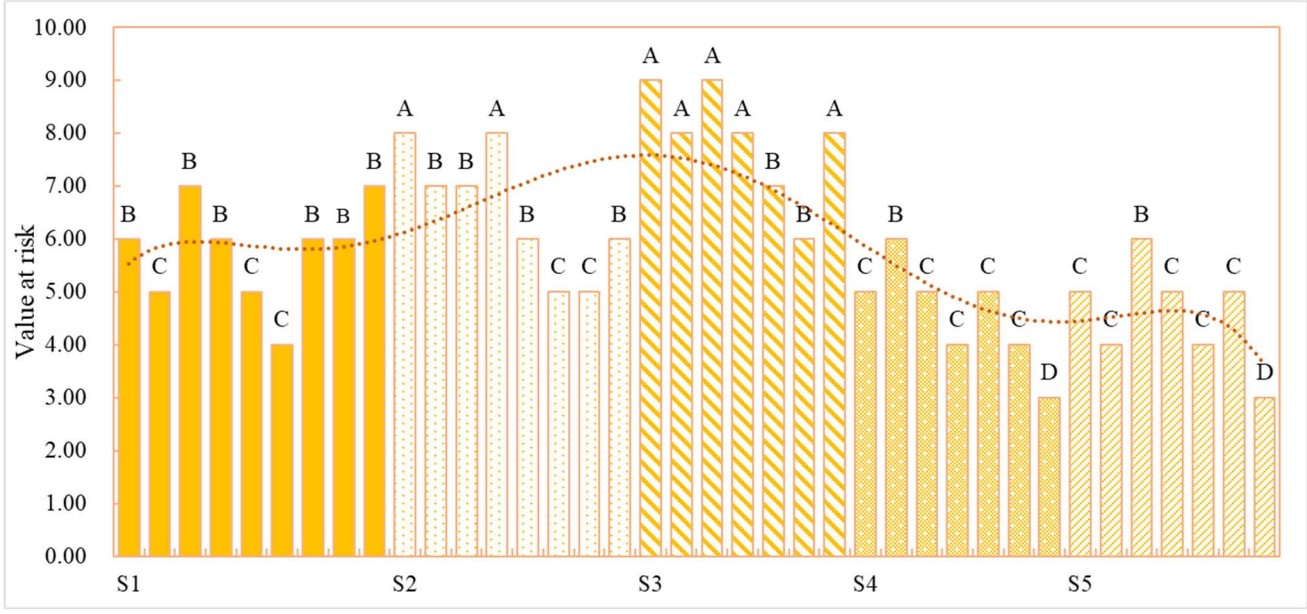

**Fig 6. Distributions of risk value and risk level.** (A, B, C, D, and E represent the risk levels from high to low, and S1, S2, S3, S4, and S5 represent the risk attributes from low to high.).

management strategies based on their specific circumstances. Large cities may need to focus more on defending against complex cyber-attacks, while smaller cities may prioritize infrastructure development and improving security awareness.

## Investigation results of new folk customs and analysis of mythical experience

A random survey of 1,140 citizens from 38 SCs identifies the typical emerging folk customs in SCs, which include offline shopping festivals, New Year's Eve celebrations, Tibet tourism, spiritual practices, green leisure, and suburban/rural tourism. The support rates for these six emerging folk customs are as follows: 17.6%, 16.7%, 15.6%, 16.2%, 16%, and 15.8%, respectively (Fig 7). Support for other options is only 2.1%. Therefore, offline shopping festivals, green leisure, and suburban/rural tourism are identified as the most popular emerging folk customs.

## Correlation between risk levels and emerging folk customs

To assess the correlation between risk levels and cultural activities, the study analyzes the distribution of emerging folk customs across cities with different risk levels (A, B, C, D, E), identifying the potential relationship between risk levels and cultural activities (Table 5). The results indicate that citizens in cities of different risk levels exhibit varying levels of support for emerging folk customs. In A-level cities (*e.g.,* Wuhan, Beijing, Shanghai), citizens show stronger support for modern activities, particularly offline shopping festivals (22.0%) and green leisure (18.2%). In contrast, B-level cities (*e.g.,* Zhuhai, Zhengzhou, Fuzhou) display more balanced support, with offline shopping festivals still popular, but traditional activities such as spiritual practices and Tibet tourism receiving less support. In C-level cities (*e.g.,* Changsha, Shenyang), citizens are more inclined to support natural activities like green leisure and suburban/rural tourism, while support for modern activities is weaker. D-level cities (*e.g.,* Huaian, Langfang) show the lowest overall support, particularly for modern activities.

This study investigates and analyzes the mythic life experiences of individuals in SCs. The survey reveals that the six most supported emerging folk customs are: offline shopping festivals (17.6%), Spring Festival celebrations (16.7%), Tibet tourism (15.6%), spiritual

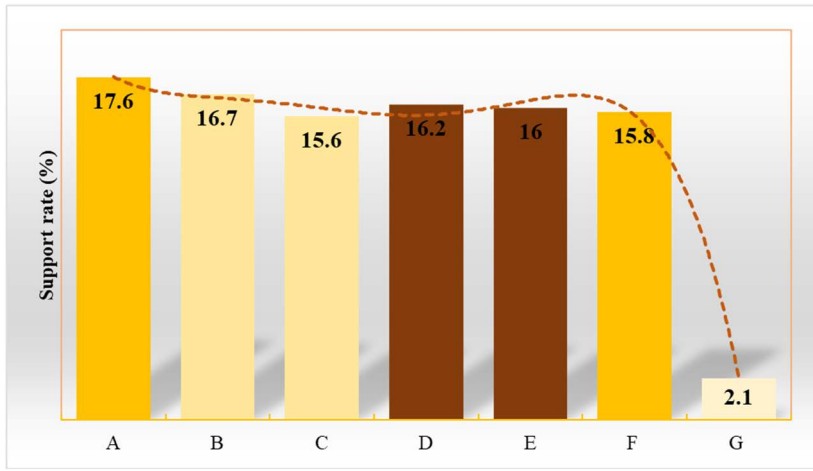

**Fig 7. Survey results of new folk customs.** (A-G represent offline shopping festivals, New Year's Eve celebrations, Tibet tourism, spiritual practices, green leisure, suburban rural tourism, and others, respectively.).

Table 5. Support rates for emerging folk customs in cities with different risk levels.

| Risk level | City | Offline Shopping Festival | New Year's Eve celebrations | Tibet Tourism | Spiritual practices | Green leisure | Suburban/rural tourism | Others |
|---|---|---|---|---|---|---|---|---|
| A | Wuhan, Nanjing, Beijing, Shenzhen, Shanghai, Guangzhou, Hangzhou | | | | | | | |
| | | 22.00% | 20.30% | 18.90% | 17.50% | 18.20% | 17.80% | 2.00% |
| B | Zhuhai, Nanchang, Zhengzhou, Fuzhou, Kunming, Dalian, Guiyang, Tangshan, Hefei, Suzhou, Chengdu, Nanjing, Qingdao, Zhangzhou, Yangzhou, Lanzhou, Tianjin, Jinan, Hangzhou, Xi'an | | | | | | | |
| | | 19.30% | 18.70% | 17.30% | 16.00% | 17.10% | 16.90% | 2.00% |
| C | Changsha, Chongqing, Shenyang, Luoyang, Harbin, Shantou, Huai'an, Wuxi, Ningbo, Xiamen, Changchun, Nanning, Langfang | | | | | | | |
| | | 16.50% | 15.80% | 14.60% | 14.00% | 15.00% | 14.90% | 2.30% |
| D | Huai'an and Langfang | | | | | | | |
| | | 14.00% | 13.50% | 12.80% | 12.50% | 13.20% | 12.00% | 2.00% |

practices (16.2%), green leisure (16%), and suburban/rural tourism (15.8%). These support rates indicate that, while the development of SCs has promoted modern activities such as shopping, it has also led to an increased preference for outdoor activities and nature-based tourism. This trend aligns with the findings of Li et al. (2022) [32]. Based on the analysis of support for emerging folk customs in cities with different risk levels, the study reveals the relationship between urban risk levels and cultural preferences. In high-risk (A-level) cities, such as Wuhan, Beijing, and Shanghai, there is a significantly higher support for modern cultural activities like offline shopping festivals and green leisure, reflecting a strong atmosphere of modernization and innovation in these cities. Additionally, support for activities that combine modern and traditional elements, such as spiritual practices and Tibet tourism, is also prominent, indicating a high level of cultural acceptance and integration in these cities. In contrast, B-level cities (*e.g.*, Zhuhai, Nanchang, Zhengzhou, and Fuzhou) show a more balanced support pattern. While offline shopping festivals still maintain a relatively high level of support, the support for spiritual practices and Tibet tourism has decreased, suggesting that these cities exhibit a lower degree of integration between modern and traditional cultures. In C-level cities (such as Changsha, Shenyang, Luoyang, Harbin), citizens show a stronger preference for nature- and leisure-related activities, such as green leisure and suburban rural tourism, while their support for modern cultural activities is relatively weaker. These cities' residents are less engaged in innovative modern activities and tend to seek more traditional and nature-oriented experiences. D-level cities (such as Huai'an, Langfang) exhibit the lowest overall support, particularly for modern activities, which may be closely related to the more pronounced traditional cultural backgrounds in these cities. These survey results further indicate that high-risk cities tend to favor modernization and innovative cultural activities, while low-risk cities show a greater interest in traditional cultural activities. This cultural disparity is closely linked to the cities' risk levels and may be influenced by factors such as cultural openness, economic development, and the lifestyle of their residents. Based on these findings, this study further analyzes the impact of these new folk activities on citizens' life experiences. The popularity of offline shopping festivals and green leisure activities indicates that SCs not only foster the development of shopping culture but also promote the popularity of green tourism and recreational activities. Meanwhile, Tibet tourism and spiritual practices reflect the citizens' desire to escape the competitive pressures of urban life and seek inner peace and spiritual solace. Overall, these new folk activities represent a fusion of modern urban culture and traditional culture, demonstrating people's pursuit of physical and mental balance as well as improved quality of life. These activities not only enrich the cultural life of cities but also reflect the combined demand for both traditional and modern cultural experiences. They provide valuable insights into the cultural development of SCs.

## Conclusions

To provide a safer development space for the development of SCs, an information security assessment model is established under KMC combined with C4.5 DT classification according to information security issues. The actual SC information risk level analysis is carried out through a questionnaire survey, to explore the accuracy and usability of the model. The folk culture in the 38 cites is also investigated through street interviews, to explore the connection between the mythical experience of people in SC and folk customs. The algorithm model constructed in this work can effectively evaluate the information security risks of SCs and has practical value. A good city image and mythical experience are driving the development of cities. The mythical experience of city image reflects people's pursuit of city development. However, the scope of empirical research in this work is limited, and the information security risk assessment indicator system in SC is not comprehensive enough. Therefore, it is necessary to enhance and improve for the further research. With this work, it needs to be realized that information security assessment has a crucial application significance in the future of SC development, which also requires more researches.

## Author contributions

**Conceptualization:** Haotong Han.

**Methodology:** Haotong Han.

**Software:** Haotong Han.

**Writing – original draft:** Haotong Han.

**Writing – review & editing:** Haotong Han.

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
