## [Decision Letter · Decision Letter 0]

1 Nov 2024

PONE-D-24-31209Adoption of K-Means Clustering Algorithm in Smart City Security: Analysis and Mythical Experience Analysis of Urban ImagePLOS ONE

Dear Dr. Han,

Thank you for submitting your manuscript to PLOS ONE. After careful consideration, we feel that it has merit but does not fully meet PLOS ONE’s publication criteria as it currently stands. Therefore, we invite you to submit a revised version of the manuscript that addresses the points raised during the review process.

We look forward to receiving your revised manuscript.

Kind regards,

Yirui Wang

Academic Editor

PLOS ONE

Additional Editor Comments:

Please revise your manuscript point by point according to reviewers' comments. Especially in experiments, evaluation metrics and dataset need to be enriched to improve the effectiveness and reliability of results and support conclusions. Besides, contributions of this manuscript need to be elaborated.

Reviewers' comments:

Reviewer's Responses to Questions

**Comments to the Author**

1. Is the manuscript technically sound, and do the data support the conclusions?

Reviewer #1: Yes

Reviewer #2: Yes

Reviewer #3: Partly

2. Has the statistical analysis been performed appropriately and rigorously? 

Reviewer #1: Yes

Reviewer #2: N/A

Reviewer #3: I Don't Know

3. Have the authors made all data underlying the findings in their manuscript fully available?

Reviewer #1: Yes

Reviewer #2: Yes

Reviewer #3: Yes

4. Is the manuscript presented in an intelligible fashion and written in standard English?

Reviewer #1: No

Reviewer #2: Yes

Reviewer #3: Yes

5. Review Comments to the Author

Reviewer #1: The comments are as follows:

1) The manuscript compares the proposed model with Naive Bayes (NB) and Logistic Regression (LR) models. Why were more advanced machine learning models, such as Random Forests, Support Vector Machines (SVM), or Gradient Boosting Machines (GBM), not included in the comparison? How might the results differ if these more sophisticated models were considered?

2) The study uses ROC curves as a performance metric, which can be misleading in cases of imbalanced datasets. How does the proposed model address potential class imbalance in the data, and why were other metrics, such as precision-recall curves or the F2 score, not considered?

3) The study uses data from 15 smart cities to validate the model. Is this sample size sufficient to generalize the findings to other smart cities globally, especially given the diverse nature of smart city implementations? How was the sample size determined, and what statistical methods were used to ensure it is representative?

4) Introduction is not focused and literature can be reorganised to strengthen literature review following contributions and discuss few relevant works i.e.,

a) Deep learning models for intelligent healthcare: implementation and challenges

b) Unsupervised pre-trained filter learning approach for efficient convolution neural network

c) CSFL: A novel unsupervised convolution neural network approach for visual pattern classification

d) Optimization of CNN through novel training strategy for visual classification problems

e) Face recognition: A novel un-supervised convolutional neural network method

5) The results indicate that cities with the same attributes exhibit minor differences in risk levels, while cities with different attributes show significant variations. Can the authors provide a more detailed explanation of what these differences in attributes entail and how they specifically contribute to variations in risk levels?

Reviewer #2: I reviewed the manuscript titled 'Adoption of K-Means Clustering Algorithm in Smart City Security: Analysis and Mythical Experience Analysis of Urban Image'. After a careful examination, I found that the manuscript is complete and well-written. The methodology used in the study is highly accurate and valuable. The research questions have been addressed effectively. The content presented in the manuscript has been meticulously collected. Based on these observations, it appears that the study can be accepted in its current form."

Reviewer #3: This paper evaluates information security in smart cities (SC) through a combined K-Means clustering and C4.5 decision tree model, examining not only security assessments but also residents' mythological experiences in these cities. The following review highlights the paper’s strengths and areas for improvement:

Strengths

Innovative Methodology: The use of a combined K-Means clustering and C4.5 decision tree for information security assessment in smart cities is a novel approach that could offer new insights into security evaluations.

Strong Results and Model Comparison: The paper reports high accuracy and an AUC greater than 0.9 for the combined model. Comparisons with other methods, such as Naïve Bayes (NB) and Logistic Regression (LR), further support the effectiveness of the combined model.

Cultural and Mythological Analysis: Integrating an analysis of cultural and mythological experiences with urban security is an original approach that adds a unique dimension, highlighting the potential relationship between culture and security in urban spaces.

Weaknesses

Lack of Clear Link Between Security and Culture: While the paper attempts to connect security and cultural experiences in smart cities, this relationship remains somewhat unclear. More detailed explanations on how information security is related to mythological experiences would add clarity.

Insufficient Details in Methodology: The methodology section is somewhat brief. For instance, further explanation on the Delphi method, the selection of experts, and other assessment steps would enhance the reader's understanding and confidence in the findings.

Need for More Comprehensive Data: The study focuses on only 15 cities, which may be insufficient for drawing broader conclusions on urban information security and cultural experiences. Expanding security assessment indicators and including additional variables would strengthen the model.

Suggestions for Improvement

Expand Methodology Details: Providing further details on data collection, standardization, and the rationale for selecting the combined K-Means and C4.5 model would give readers a clearer understanding of the model's effectiveness.

Increase Sample Size and Data Variety: Expanding the study to cover more cities and examine additional indicators would enhance the generalizability of the results.

Clarify the Link Between Security and Culture: Clearer explanations on how information security impacts cultural and spiritual experiences of city residents could enhance the paper’s value and originality.

Conclusion

Overall, the paper is logically structured and presents an innovative approach, yet it could benefit from additional detail and a stronger conceptual link between security and culture. Addressing these areas would allow the paper to become a valuable resource in assessing the relationship between information security and cultural experience in smart cities.

6. PLOS authors have the option to publish the peer review history of their article (what does this mean? ). If published, this will include your full peer review and any attached files.

**Do you want your identity to be public for this peer review?** For information about this choice, including consent withdrawal, please see our Privacy Policy .

Reviewer #1: No

Reviewer #2: **Yes**

Reviewer #3: No

---

## [Author Response · Author response to Decision Letter 0]

27 Nov 2024

We would like to express our sincere gratitude to you and the reviewers for your thoughtful evaluation of our manuscript and for the invaluable feedback provided. We highly appreciate each suggestion and have carefully revised the manuscript in accordance with the reviewers' and editors' comments. We are now submitting the revised manuscript for your further consideration.

---

## [Decision Letter · Decision Letter 1]

6 Jan 2025

PONE-D-24-31209R1Adoption of K-Means Clustering Algorithm in Smart City Security: Analysis and Mythical Experience Analysis of Urban ImagePLOS ONE

Dear Dr. Han,

Thank you for submitting your manuscript to PLOS ONE. After careful consideration, we feel that it has merit but does not fully meet PLOS ONE’s publication criteria as it currently stands. Therefore, we invite you to submit a revised version of the manuscript that addresses the points raised during the review process.

We look forward to receiving your revised manuscript.

Kind regards,

Yirui Wang

Academic Editor

PLOS ONE

Additional Editor Comments:

Please further improve your manuscript according to new reviewer's comments.

Reviewers' comments:

Reviewer's Responses to Questions

**Comments to the Author**

1. If the authors have adequately addressed your comments raised in a previous round of review and you feel that this manuscript is now acceptable for publication, you may indicate that here to bypass the “Comments to the Author” section, enter your conflict of interest statement in the “Confidential to Editor” section, and submit your "Accept" recommendation.

Reviewer #2: All comments have been addressed

Reviewer #4: All comments have been addressed

2. Is the manuscript technically sound, and do the data support the conclusions?

Reviewer #2: Yes

Reviewer #4: Yes

3. Has the statistical analysis been performed appropriately and rigorously? 

Reviewer #2: Yes

Reviewer #4: Yes

4. Have the authors made all data underlying the findings in their manuscript fully available?

Reviewer #2: Yes

Reviewer #4: Yes

5. Is the manuscript presented in an intelligible fashion and written in standard English?

Reviewer #2: Yes

Reviewer #4: Yes

6. Review Comments to the Author

Reviewer #2: I checked the revised version. I noticed that you answered all the revisions correctly. Now it seems that this manuscript has reached an acceptable level in terms of quality.

Reviewer #4: Comments on the manuscript entitled

“Adoption of K-Means Clustering Algorithm in Smart City Security: Analysis and Mythical Experience Analysis of Urban Image”

The manuscript presents a well-structured and coherent proposal for a hybrid model, demonstrating an improvement in accuracy over traditional models such as logistic regression and Naive Bayes. The authors have adequately addressed the reviewers' queries, indicating a thorough understanding of the research domain. In my opinion, it can be published in the journal after addressing the following comments/suggestions.

Some Comments/ Recommendations

1. While the model's performance is superior to simpler models e.g. LR and NB, it does not exhibit a significant improvement over more complex models like SVM and Random Forest. It would be beneficial for the author(s) to explore and discuss the possible level of interpretability of their proposed model in relation to those of counterparts mentioned.

2. First, there is no explanation on how the data has been splitted to train/test sets. Then, given the relatively small number of observations, it's essential to assess whether the models, particularly the new one, suffer from overfitting or underfitting or not.

Additional Remarks

• The formulae, specifically (1), (2), and (7), could be rewritten for clarity and consistency, potentially following the format of (8).

• Line 315: Simplify the notation to "F1-F20" for better readability.

• Eq (4): distance

7. PLOS authors have the option to publish the peer review history of their article (what does this mean? ). If published, this will include your full peer review and any attached files.

**Do you want your identity to be public for this peer review?** For information about this choice, including consent withdrawal, please see our Privacy Policy .

Reviewer #2: **Yes:**

Reviewer #4: **Yes: **

---

## [Author Response · Author response to Decision Letter 1]

15 Jan 2025

Dear Editor and Reviewers：

Thank you and the reviewers for your valuable comments on our manuscript Adoption of K-Means Clustering Algorithm in Smart City Security: Analysis and Mythical Experience Analysis of Urban Image. We have carefully read the reviewers’ comments and made corresponding modifications and improvements to the manuscript according to the feedback. We now submit the revised manuscript to you, along with the following contents:

1. Response Letter: We have replied to each comment raised by the academic editor and reviewers, and have provided detailed explanations of the modifications made in the text. The response letter has been uploaded, with the file name “Response to Reviewers”.

2. Revised Manuscript with Track Changes: We have uploaded a copy of the manuscript with track changes, so that the reviewers can clearly find all the modifications we have made to the article. The file name is “Revised Manuscript with Track Changes”.

3. Revised Manuscript without Track Changes: In addition, we have also uploaded a copy of the manuscript without track changes. The file name is “Manuscript”.

We have further optimized and explained the experimental design according to the reviewers’ suggestions.

In addition, storing the laboratory experimental protocol on the protocols.io platform is not applicable.

Thank you again for your recognition of our research and for the careful review of the manuscript. We look forward to your further feedback. 

6. Review Comments to the Author

Reviewer #2: I checked the revised version. I noticed that you answered all the revisions correctly. Now it seems that this manuscript has reached an acceptable level in terms of quality.

Reviewer #4: Comments on the manuscript entitled

“Adoption of K-Means Clustering Algorithm in Smart City Security: Analysis and Mythical Experience Analysis of Urban Image”

The manuscript presents a well-structured and coherent proposal for a hybrid model, demonstrating an improvement in accuracy over traditional models such as logistic regression and Naive Bayes. The authors have adequately addressed the reviewers' queries, indicating a thorough understanding of the research domain. In my opinion, it can be published in the journal after addressing the following comments/suggestions.

Some Comments/ Recommendations

1. While the model's performance is superior to simpler models e.g. LR and NB, it does not exhibit a significant improvement over more complex models like SVM and Random Forest. It would be beneficial for the author(s) to explore and discuss the possible level of interpretability of their proposed model in relation to those of counterparts mentioned.

Reply: We appreciate the reviewer’s suggestions. While the C4.5 decision tree model based on KMC performs excellently when compared with simpler models such as LR and NB, the performance improvement is not significant when compared with more complex models like SVM and random forest. To better understand and apply this model, we will further analyze its interpretability in the discussion. Specifically, the tree structure of the C4.5 decision tree model is easy to understand and interpret, which gives it an advantage in transparency and interpretability of the decision-making process compared to models like SVM and random forest. We will further discuss this point and provide a comparison of interpretability with models such as SVM and random forest.

2. First, there is no explanation on how the data has been splitted to train/test sets. Then, given the relatively small number of observations, it's essential to assess whether the models, particularly the new one, suffer from overfitting or underfitting or not.

Reply: We agree with the reviewer’s concern about the division of the training set/test set. We will clarify the method of dataset division in the revision, adopting the common 10-fold cross-validation method to divide the data to ensure the stability and generalization ability of the model. In addition, due to the small number of observations, we have added an assessment of overfitting and underfitting in the results section.

Additional Remarks

• The formulae, specifically (1), (2), and (7), could be rewritten for clarity and consistency, potentially following the format of (8).

Reply: We appreciate the reviewer’s suggestions regarding the clarity and consistency of the formulas. In the revision, we will rewrite formulas (1), (2), and (7) according to the reviewer’s suggestions to ensure that they are expressed more clearly and in accordance with the principle of consistency. Specifically, we will refer to the format of formula (8) and make the layout of the formulas more intuitive, ensuring that each variable and operation in the formulas clearly and explicitly conveys its mathematical meaning.

• Line 315: Simplify the notation to "F1-F20" for better readability.

Reply: We agree with the reviewer’s suggestion to simplify the notation “F1-F20”. In the revision, we will directly use “F1-F20” to simplify the description, making it more concise and easier to understand. This will enhance the readability of the article and avoid redundant expressions.

• Eq (4): distance

Reply: Regarding the issue of distance calculation mentioned in formula (4), we have clarified and rewritten the formula to ensure its consistency with the model’s computation.

---

## [Decision Letter · Decision Letter 2]

5 Feb 2025

Adoption of K-Means Clustering Algorithm in Smart City Security: Analysis and Mythical Experience Analysis of Urban Image

PONE-D-24-31209R2

Dear Dr. Han,

We’re pleased to inform you that your manuscript has been judged scientifically suitable for publication and will be formally accepted for publication once it meets all outstanding technical requirements.

Kind regards,

Yirui Wang

Academic Editor

PLOS ONE

Additional Editor Comments (optional):

Reviewers' comments:

Reviewer's Responses to Questions

**Comments to the Author**

1. If the authors have adequately addressed your comments raised in a previous round of review and you feel that this manuscript is now acceptable for publication, you may indicate that here to bypass the “Comments to the Author” section, enter your conflict of interest statement in the “Confidential to Editor” section, and submit your "Accept" recommendation.

Reviewer #2: (No Response)

Reviewer #4: All comments have been addressed

2. Is the manuscript technically sound, and do the data support the conclusions?

Reviewer #2: (No Response)

Reviewer #4: Yes

3. Has the statistical analysis been performed appropriately and rigorously? 

Reviewer #2: (No Response)

Reviewer #4: Yes

4. Have the authors made all data underlying the findings in their manuscript fully available?

Reviewer #2: (No Response)

Reviewer #4: Yes

5. Is the manuscript presented in an intelligible fashion and written in standard English?

Reviewer #2: (No Response)

Reviewer #4: Yes

6. Review Comments to the Author

Reviewer #2: With these revisions, the quality of the manuscript has been enhanced, and it is now accepted.

Reviewer #4: (No Response)

7. PLOS authors have the option to publish the peer review history of their article (what does this mean? ). If published, this will include your full peer review and any attached files.

**Do you want your identity to be public for this peer review?** For information about this choice, including consent withdrawal, please see our Privacy Policy .

Reviewer #2: **Yes**

Reviewer #4: No

---

## [Editor Report · Acceptance letter]

PONE-D-24-31209R2

PLOS ONE

Dear Dr. Han,

I'm pleased to inform you that your manuscript has been deemed suitable for publication in PLOS ONE. Congratulations! Your manuscript is now being handed over to our production team.

Kind regards,

on behalf of

Dr. Yirui Wang

Academic Editor

PLOS ONE